# Effective Coverage of Maternal and Newborn Health Services in Sub-Saharan Africa: What distinguishes high from medium and low performers?

Ayelign Mengesha Kassie[1,2]*, Solomon Woldeyohannes[3,4], Anteneh Zewdie[5], Eskinder Wolka[5], Yibeltal Assefa[1]

**1** School of Public Health, Faculty of Health, Medicine and Behavioral Science, The University of Queensland, Brisbane, Australia, **2** School of Nursing, College of Health Sciences, Woldia University, Woldia, Ethiopia, **3** School of Veterinary Science, The University of Queensland, Brisbane, Australia, **4** School of Translational Medicine, Faculty of Medicine, Nursing and Health Sciences, Monash University, Melbourne, Australia, **5** International Institute for Primary Health Care-Ethiopia, Addis Ababa, Ethiopia

☉ These authors contributed equally to this work.
* ayelignmengesha59@gmail.com

## Abstract

### Background

Effective coverage (EC) has emerged as a better measure of service coverage, in the past decades, compared to the simple crude coverage measures. It represents the proportion of a population in need of a service that successfully receives it with sufficient quality to achieve the intended health benefits. Nevertheless, EC in maternal and newborn health (MNH) services are significantly variable across and within countries. Therefore, this study aimed to identify the societal and health system factors that can explain why some countries are having higher EC of MNH services than others in Sub-Saharan Africa (SSA).

### Methods

A mixed-method case study design was employed with inclusion of document review. Effective coverage rates were estimated using countries demographic and health survey (DHS) datasets. Two countries were then selected for each MNH service domain from each performance category, high, medium, and low, for further analysis of explanatory factors. Data sources included DHS and health facility survey summary reports, the Global Health Expenditure Database, and TheGlobalEconomy.com.

### Results

We found huge variation in EC of MNH services across countries in SSA. The scores range from 7% in Ethiopia to 64% in Liberia for 4+ ANC visits, 9% in Ethiopia and Nigeria to 81% in Rwanda for institutional delivery, 3% in Ethiopia to 77% in

which permits unrestricted use, distribution, and reproduction in any medium, provided the original author and source are credited.

**Data availability statement:** The country-level DHS microdata used in this study are available from the DHS Program upon request (https://dhsprogram.com/data/). Access requires registration and approval, and the authors are not permitted to redistribute these individual-level datasets. However, additional publicly available data extracted from DHS summary reports, Health Facility Survey reports, and global databases are provided in Additional File 2. We have also provided an Excel file as an additional file 3 item containing only aggregated indicators, graphs, and summary results derived from DHS microdata and document reviews; no individual-level DHS data are included.

**Funding:** The author(s) received no specific funding for this work.

**Competing interests:** The authors declare no competing interests.

**Abbreviations:** ANC, Antenatal Care; EC, Effective Coverage; BEmONC, Basic Emergency Obstetric & Newborn Care; DHS, Demographic and Health Survey; MNH, Maternal and Newborn Health; PNC, Postnatal Care; SSA, Sub-Saharan Africa; TFR, Total Fertility Rate.

Gambia for PNC mothers, and 1% in Ethiopia to 68% in South Africa for PNC newborns. These discrepancies are highly likely influenced by multilevel health system and societal factors. High-performing countries in EC of MNH services have higher service availability and readiness scores than medium- and low-performing ones. For instance, Ghana and Liberia scored 83% and 84%, respectively, for tracer indicators of ANC service availability, compared to 43% in Ethiopia and 64% in Malawi. Similar pattern is observed between the selected countries EC estimates of MNH services and their health service specific readiness index scores. In addition, they also have favourable societal factors including high proportion of women attending primary and/ or more school levels, better mass media and internet access, and relatively lower political instability indexes. Low-performing countries like Ethiopia and Nigeria had complex futures including having low health service availability and readiness scores and unfavourable societal factors including in women's education, and internet and mass media access. Furthermore, the two countries had weakest average political stability index that hinders the utilization and delivery of MNH services.

## Conclusions

The findings revealed that better health service availability and readiness, strong healthcare financing, favourable societal factors and having a relatively stable political index are critical in determining countries performance in EC of MNH services. Therefore, countries, particularly low performers in EC of MNH services need to learn from positive outliers in improving EC of MNH services. Strengthening existing health facilities with better staffing, training, and resources is crucial beyond merely expanding new ones.

## Background

Despite progress in healthcare access and coverage, maternal and newborn mortality remains high, particularly in resource-limited settings [1]. In 2022, 2.3 million newborns died within their first month of life, with Sub-Saharan Africa and South Asia recording the highest neonatal mortality rates at 27 and 21 deaths per 1000 live births, respectively [2]. Maternal mortality is similarly concerning, with 94% of deaths occurring in low-resource settings with majority of them due to preventable causes such as severe bleeding, infections, and unsafe abortions [3].

While healthcare service coverage, including antenatal care and immunization, has improved globally [4], researchers claim that high coverage alone cannot significantly reduce mortality due to quality gaps [5,6]. This has resulted in the introduction of the concept of "effective coverage" (EC) which integrates access, quality, and health outcomes to provide a more accurate assessment of healthcare systems [7]. Effective coverage adjusts the quality and appropriateness of the care provided to contact coverage rates, ensuring that the services delivered meet the standards necessary to achieve intended health outcomes [7,8].

In maternal and newborn health (MNH) care, EC estimates are calculated by adjusting facility visits with the services provided, measuring both access and intervention effectiveness [7,9]. Significant gaps exist in EC rates of MNH services across and within countries [10]. For instance, Hodgins S et al. conducted a study using Demographic and Health Survey (DHS) data from 41 countries to assess coverage for specific elements of antenatal care (ANC). In this study, EC was measured by calculating the simple average of a set of available indicators for the receipt of specific services, which served as a summary measure of antenatal program performance at the population level. The study reported that EC of ANC services ranges from 22% in Ethiopia to 84% in the Dominican Republic [11].

Nevertheless, there is limited evidence about the reasons why variations occur in EC of MNH services across and within countries [12,13]. Most of the studies that are conducted to identify the factors affecting EC focus on individual and/ or lower community structure level factors including household decision-making autonomy and husbands influence at family levels. However, the factors influencing EC of MNH services are complex and operate at multiple levels, ranging from individual to societal and policy-related factors [14]. For instance, at organization level, health facility capacity constraints, including shortages of human resources, equipment, diagnostics, medicines, and other essential commodities, can negatively affect EC by diminishing the quality of care provided, despite women attending health facilities [15–17].

Furthermore, socio-demographic factors impact EC not only by shaping health-seeking behaviours and reducing contact coverage rates but also by exacerbating disparities in the quality of care provided at health facilities, unlike the health facility capacity related factors. For example, Fink et al. reported that women in the wealthy group tend to receive higher-quality care than their counterparts, despite visiting the same health facilities. The study suggested several reasons for this disparity, including wealthier women's ability to pay additional fees for services such as consultations, diagnostic tests, and consumables [18]. Therefore, this study aims to identify the societal and health system factors that can explain the variations in EC of MNH care services in Sub-Saharan Africa using a case study approach. We believe that this approach can help low-and medium performing countries in EC of MNH services in Sub-Saharan Africa learn from the success of positive deviants (high performing countries).

## Conceptual framework

As the factors influencing EC of MNH services are complex and operating at multiple levels, using the socio-ecological model to depict the relationship between explanatory and outcome variables is paramount [14]. However, we have not included individual level factors in our analysis as our analysis is focusing on health system and aggregate community/ societal level factors (**Fig 1**).

## Methods

### Study setting, design and sampling procedure

A mixed-method case study has been employed to identify the societal and health system factors that can explain variations in EC of MNH services in Sub-Saharan Africa. For this objective, a combination of countries Demographic and Health Survey (DHS) and their summary report, and data from other sources including the countries service availability and readiness assessment surveys were utilized. The DHS uses stratified sampling procedures to identify representative samples for countries [22]. In this study we used a weighted sample of 118,614 reproductive age group women who have delivered a live newborn in the 2 years prior to 27 SSA country DHS surveys. We based our selection on the 2 years as data collection regarding MNH services in those surveys is different across the spectrum. For instance, only women who had live births in the 2 years before the DHS were considered for PNC service delivery data collections in several countries including Ethiopia [23,24]. Accordingly, we analysed routine MNH visits from the antenatal to postnatal periods, and EC rates were estimated for each of these domains. These estimates served as the foundation for the selection of cases to further analyse the societal and health system factors that can explain the variations in the EC of MNH services across countries.

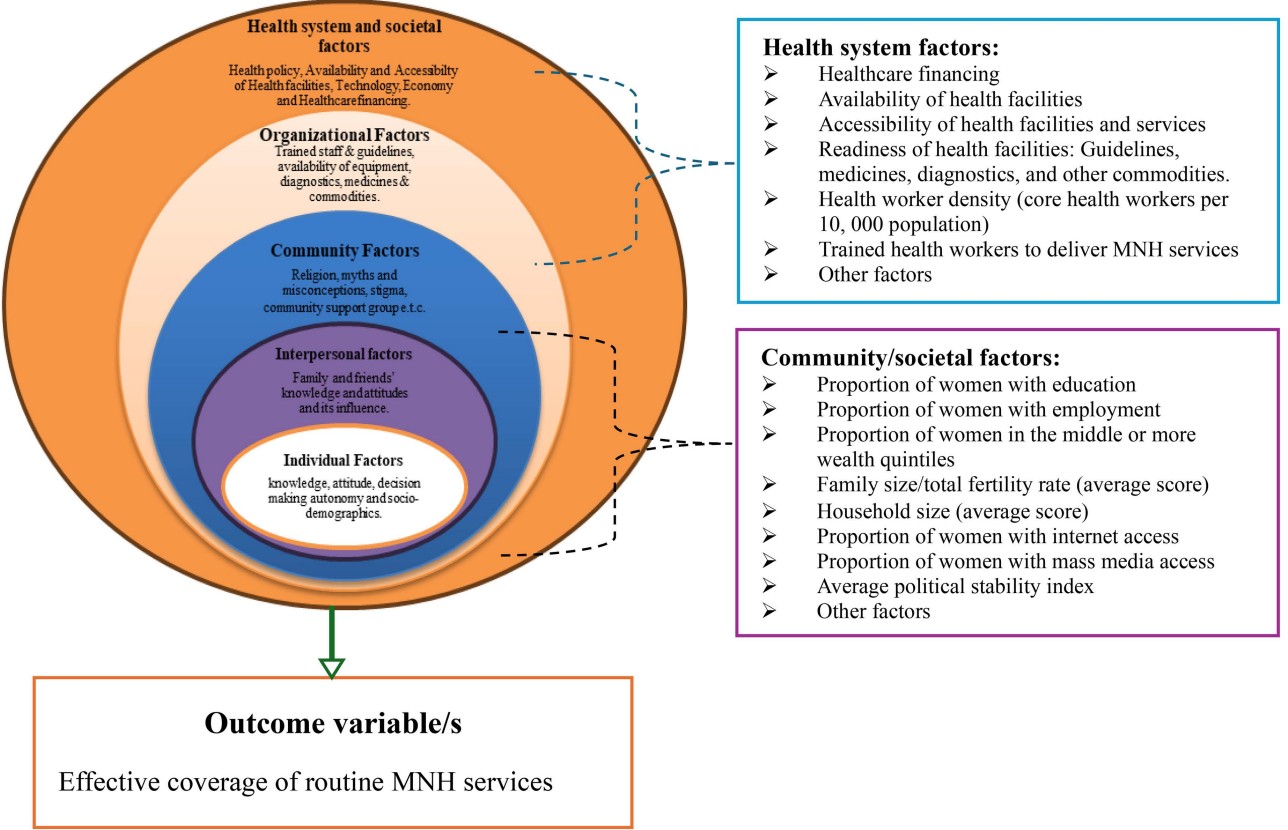

**Fig 1. The socio-ecological model depicting multilevel factors that can affect EC of routine MNH services, adapted from previous literature [15–17,19–21].**

## Selection of cases for comparison and document analysis

The case study approach was employed to compare high-, medium-, and low-performing countries in Sub-Saharan Africa based on their relative performance. As such, EC rates of routine MNH services estimated based on the countries DHS data were considered. Countries were categorized into three groups based on their EC estimates for routine MNH services. Subsequently, two high-performing countries (positive outliers), two medium-performing countries, and two low-performing countries (negative outliers) were selected for each domain. Some countries, such as Ethiopia, were selected repeatedly due to their consistent performance ranking across the spectrum of services, with Ethiopia ranking lowest in all domains. Then, we utilized the countries health facility survey data sources to compare the selected countries health service availability and delivery capacity indexes [25–30]. On the other hand, countries DHS summary reports, Global Health Expenditure Database and TheGlobalEconomic.com websites were utilized to identify community and/or societal-level factors that could explain the variation in performance among those countries [23,31–40].

## Operational definition

**Outcome variable/s. Intervention coverage.** Intervention coverage was estimated based on the average quality score of MNH service indicators at the population level. Ten [10] items were utilised to estimate the average intervention coverage score for 4[+]ANC visits except for Mauritania and South Africa in which 9 items were used due to one indicator that is receipt of intestinal parasitic drugs during pregnancy being missing from the selected 10 indicators of service

delivery quality in the dataset. In addition, 4 items for institutional delivery, 3 items for PNC for mothers within two days post-delivery and 8 items for PNC for newborns within two days post-delivery were considered to estimate service delivery quality for each woman and their newborns. Then, an overall average score was estimated for each of the domains at the population level. To do these, 0 value was given for women who did not visit health facilities and/or did not fulfil the required health facility visits according to the WHO standards. For those women who fulfilled the required visits, the average quality score was estimated based on the total number of indicator items, for each woman (**See Table 2 in** S1 File).

### Effective coverage

Effective coverage is a composite measure of service coverage and is determined by combining contact coverage and intervention coverage rates (average service quality scores). It is calculated using the formula $EC_{ij} = \sum Q_{ij} \times U_{ij} | N_{ij} = 1$, where $EC_{ij}$ represents the effective coverage for an individual i receiving intervention j. Here, Q denotes the proportion of potential health gain achieved through the intervention, which in this context equates to the average quality score of healthcare services received by women [41]. U refers to the probability of receiving the intervention/utilization of services, conditional on need, and is represented as contact coverage rates for specific interventions/services such as attending $4^+$ANC visits [42]. For example, if contact coverage is 100%, it means all women in need of the service attended at least 4 ANC visits. Intervention coverage on the other hand reflects the quality of the services provided, indicating how well the services met the required standards. If the population average intervention coverage is 50%, then, EC will be 50%. This means that, on average, 50% of women who attended $4^+$ANC visits have received all the required services according to the standards. When both contact and intervention coverage are not perfect, the EC will reflect the combined effect of these two factors, but the gap between coverage and EC will not always be halved. Instead, it will be proportional to both the intervention coverage score and the contact coverage rate. For instance, in Ethiopia's case, where contact coverage for $4^+$ANC visits is 33.34% and average intervention coverage is 21.1%, the EC becomes 7.03%. This means that only 7.03% of women both attended $4^+$ANC visits and received all the required services according to standards, on average (**See** S1 File).

**Explanatory variables. Health system level factors.** Health system level factors are assessed using health care financing, service availability and readiness scores for the selected countries in each category by using the countries health facility survey summary reports and other sources of data. Availability is used to refer the percentage of health facilities offering specific services and the presence of tracer items for different inputs, such as the availability of diagnostic, essential medicines, and infrastructure resources across the health facilities in those countries. For instance, an average index for the availability of iron supplementation, tetanus toxoid vaccination, folic acid supplementation and monitoring for hypertensive disorders of pregnancy was utilised to compare countries for availability of ANC services, in addition to the proportion of health facilities that provide those services in each country. Readiness, on the other hand, is a composite measure calculated for the facilities that provide the service (restricted to the subset of facilities that offered the specific service). The components of the readiness score vary depending on the service but generally include domains such as key staff members with essential trainings, equipment, medicines and supplies, and diagnostics. A readiness score of 50 indicates that, on average, half of the facilities offering the service had all the necessary inputs for service delivery. For healthcare financing, we used an average of the 2 years before the DHS survey of countries current health expenditure per capita data (**See** S2 File) [43].

### Community and/or societal level factors

Education, employment, family size and some other important variables including media access and political stability indexes were analysed with their average scores representing the community or society of the included countries. The scores are based on the countries DHS summary reports and TheGlobalEconomic.com website. Education is used to

refer the proportion of reproductive age women with some primary or more school attendance. Employment measures the percentage of reproductive age women who were employed in the 12 months preceding the countries DHS surveys. In addition, family size is represented by the total fertility rate and household sizes which could serve as a proxy measure to reflect the impact of family size on women's health-seeking behaviours. Internet access is defined by the proportion of women who used the internet in the last 12 months preceding the surveys. Mass media access also reflects those women's access to at least one of the three media sources (newspapers, television, or radio) at least once a week [23,31–39]. On the other hand, political stability index is a composite measure derived from multiple sources, including the Economist Intelligence Unit, the World Economic Forum, and Political Risk Services. It assesses the likelihood of disruptions such as undemocratic transfers of power, armed conflicts, violent protests, social unrest, international disputes, terrorism, and ethnic or regional tensions. The index, available from 1996 to 2023, is measured on a continuum scale (−2.5 weak; 2.5 strong), where negative values indicate weak political stability, and positive values signify strong stability. The consistency of the index over time allows for meaningful comparisons across different periods and countries. As such, we have used the average score of the indicated period in our analysis [40] (**See** S2 File).

### Ethical approval and consent to participate

Secondary data sources were used in this study with data access permission obtained from the DHS Program for using countries' DHS datasets. The other articles included for document review do not require formal ethical approval as they are publicly available and contain no personally identifiable information.

### Data analysis and presentation

We have used countries DHS data covering the period of 2015−16–2022−23. These data were analysed descriptively to estimate coverages of 4+ANC, institutional delivery, first PNC visit for mothers, and first PNC visits for newborns within two days and the EC rates of those visits. We used Stata Software Version 18.0 to analyse those data. All analyses is conducted using the 'svy' command function and considering the clustering effect of the complex sample design used in DHS [44]. All reported estimates were weighted (unless otherwise indicated). In addition, the explanatory factors extracted from the countries DHS and health facility survey summary reports, Global Health Expenditure Database, Global Health Observatory data repository and TheGlobalEconomic.com website were analysed manually using the excel sheet and the results are presented in text, tables and graphs.

## Results

### Background characteristics

We found significant disparities in MNH service utilization among women who had live births in the two years preceding the DHS of included SSA countries. The coverage of 4+ ANC visits ranges from 33% in Ethiopia to 87% in Liberia and Ghana. The contact coverage rate for institutional delivery also ranges from 36% in Ethiopia to 96% in South Africa. Moreover, PNC coverage within 2 days ranges from 17% in Ethiopia to 88% in Gambia, for mothers, and from 13% in Ethiopia to 87% in Ghana and South Africa, for newborns. Ethiopia has the lowest rate across all domains. These rates are deeply concerning given Ethiopia's persistently high maternal and neonatal mortality rates (Table 1).

### Intervention Coverage of MNH Services in Sub-Saharan Africa

The average intervention coverage rates for 4+ANC visits, institutional delivery, PNC for mothers, and PNC for newborns were around 44%, 53%, 55% and 42%, respectively, in SSA. The scores range from 21% in Ethiopia to 73% in Liberia for 4+ANC services. Similarly, intervention coverage for institutional delivery varied between 22% in Nigeria and 86% in

**Table 1. Contact coverage of routine MNH visits in Sub-Saharan Africa (N = 118,614).**

| Country | Year of Survey | Frequency, N | 4⁺ANC visits, n (%) | Institutional delivery, n (%) | PNC visit within 2 days mothers, n (%) | PNC visit within 2 days newborns, n (%) |
|---|---|---|---|---|---|---|
| Angola | 2015−16 | 5,405 | 3,226 (59.68) | 2,551(47.19) | 1,297 (23.99) | 1,138 (21.05) |
| Benin | 2017−18 | 5,502 | 2,785 (50.61) | 4,680 (85.06) | 3,648 (66.30) | 3,555 (64.62) |
| Burkina Faso | 2021 | 4,684 | 3,369 (71.92) | 4,399 (93.93) | 3,744 (79.93) | 3,715 (79.31) |
| Burundi | 2016−17 | 5,412 | 2,799 (51.71) | 4,620 (85.37) | 2,765 (51.09) | 2,680 (49.51) |
| Cameroon | 2018 | 3,924 | 2,485 (63.33) | 2,657 (67.72) | 2,353 (59.98) | 2,395 (61.04) |
| Cote d'Ivoire | 2021 | 3,858 | 2,154 (55.83) | 3,092 (80.15) | 2,857 (74.04) | 2,799 (72.54) |
| Ethiopia | 2015−16 | 4,308 | 1,436 (33.34) | 1,560(36.22) | 715 (16.61) | 575 (13.34) |
| Gabon | 2019−21 | 2,456 | 1,910 (77.75) | 2,332 (94.93) | 1,816 (73.94) | 1,950 (79.37) |
| Gambia | 2019−20 | 3,129 | 2,481 (79.28) | 2,712 (86.66) | 2,762 (88.27) | 2,614 (83.52) |
| Ghana | 2022 | 3,491 | 3,034 (86.90) | 2978 (85.30) | 3,052 (87.41) | 3,020 (86.51) |
| Guinea | 2018 | 3,026 | 1,081 (35.72) | 1,640 (54.21) | 1,499 (49.54) | 1,316 (43.48) |
| Kenya | 2022 | 6,847 | 4,495 (65.65) | 6,006 (87.71) | 5,335 (77.92) | 5,659 (82.64) |
| Liberia | 2019−20 | 2,096 | 1,817 (86.72) | 1,744 (83.22) | 1,678 (80.05) | 1,596 (76.16) |
| Madagascar | 2021 | 4,897 | 2,843 (58.05) | 1,919 (39.19) | 2,797 (57.11) | 2,221 (45.35) |
| Malawi | 2015−16 | 6,693 | 3,221 (48.13) | 6,214 (92.86) | 2,848 (42.55) | 3,995 (59.69) |
| Mali | 2018 | 4,150 | 1,791 (43.15) | 2,899 (69.86) | 2,399 (57.82) | 2,279 (54.91) |
| Mauritania | 2019−21 | 4,485 | 1,762 (39.29) | 3,258 (72.64) | 1,938 (43.21) | 1,804 (40.23) |
| Mozambique | 2022−23 | 3,822 | 1,853 (48.47) | 2,452 (64.16) | 1,452 (37.98) | 1,587 (41.52) |
| Nigeria | 2018 | 12,935 | 7,267 (56.18) | 5,248 (40.57) | 5,512 (42.61) | 4,932 (38.13) |
| Rwanda | 2019−20 | 3,236 | 1,527 (47.20) | 3,042 (94.03) | 2,279 (70.43) | 2,421 (74.83) |
| Senegal | 2019 | 2,327 | 1,260 (54.14) | 1,934 (83.10) | 1,882 (80.86) | 1,917 (82.37) |
| Sierra Leone | 2019 | 3,950 | 3,138 (79.44) | 3,370 (85.32) | 3,438 (87.03) | 3,271 (82.80) |
| South Africa | 2016 | 1,386 | 1,039 (74.97) | 1,332 (96.08) | 1,168 (84.27) | 1,206 (87.00) |
| Tanzania | 2022 | 4,335 | 2,806 (64.74) | 3,504 (80.84) | 2,192 (50.57) | 2,337 (53.91) |
| Uganda | 2016 | 5,901 | 3,556 (60.25) | 4,511 (76.44) | 3,245 (54.99) | 3,316 (56.18) |
| Zambia | 2018 | 3,905 | 2,461 (63.03) | 3,370 (86.30) | 2,751 (70.44) | 2,830 (72.47) |
| Zimbabwe | 2015 | 2,454 | 1,797 (73.25) | 1,987 (80.98) | 1,398 (56.98) | 1,810 (73.76) |
| **Total** | Weighted | 118,614 | 69,392 (58.50) | 86,012 (72.51) | 68,819 (58.02) | 68,934 (58.12) |

Rwanda. Likewise, intervention coverage for PNC ranged from 16% in Ethiopia to 87% in Gambia for mothers, and from 9% in Ethiopia to 78% in South Africa for newborns (Table 2).

### Effective Coverage of MNH Services in Sub-Saharan Africa

The analysis of EC rates for MNH services across SSA regions showed significant disparities. Ghana and Liberia are the highest performers in EC rates for 4⁺ANC visits with 62.3% and 63.6% scores, respectively. Rwanda had the highest score at 80.8% for institutional delivery. Gambia and South Africa are the top scorers for PNC of mothers and newborns, respectively, with 76.7% and 67.9% EC rates. Sierra Leone, Gabon and Burkina Faso are additional examples among the countries that need recognition in achieving high performance in EC of MNH services. In contrast, some countries like Nigeria and Ethiopia demonstrated the lowest EC rates across these domains. Ethiopia is the lowest performer in EC of ANC and PNC services with 7% score for 4⁺ANC visits, 9.1% for institutional delivery (as Nigeria), 2.7% for PNC mothers,

**Table 2. Average intervention coverage (quality) score of routine MNH services in Sub-Saharan Africa (N = 118,614).**

| Country | Average intervention coverage/Quality score of… | | | | |
|---|---|---|---|---|---|
| | Frequency, N | 4⁺ANC visits (%) | Institutional delivery (%) | PNC mothers (%) | PNC newborns (%) |
| Angola | 5,405 | 47.2 | 32.8 | 23.7 | 14.6 |
| Benin | 5,502 | 42.8 | 67.2 | 65.5 | 45.1 |
| Burkina Faso | 4,684 | 57.0 | 79.6 | 79.2 | 54.5 |
| Burundi | 5,412 | 31.3 | 58.4 | 50.7 | 21.2 |
| Cameroon | 3,924 | 50.3 | 46.5 | 56.7 | 45.3 |
| Cote d'Ivoire | 3,858 | 44.2 | 49.2 | 70.5 | 40.4 |
| Ethiopia | 4,308 | 21.0 | 25.0 | 16.2 | 9.3 |
| Gabon | 2,456 | 66.9 | 81.7 | 73.3 | 67.7 |
| Gambia | 3,129 | 60.7 | 47.8 | 86.9 | 61.8 |
| Ghana | 3,491 | 71.7 | 64.5 | 84.5 | 73.9 |
| Guinea | 3,026 | 27.2 | 26.6 | 43.6 | 30.7 |
| Kenya | 6,847 | 48.5 | 55.3 | 75.9 | 51.1 |
| Liberia | 2,096 | 73.3 | 62.0 | 74.8 | 55.0 |
| Madagascar | 4,897 | 39.9 | 27.5 | 43.8 | 30.2 |
| Malawi | 6,693 | 34.4 | 78.1 | 42.1 | 52.6 |
| Mali | 4,150 | 33.4 | 40.0 | 53.4 | 33.2 |
| Mauritania | 4,485 | 30.9 | 45.8 | 42.3 | 24.4 |
| Mozambique | 3,822 | 34.1 | 51.1 | 35.4 | 29.9 |
| Nigeria | 12,935 | 40.2 | 22.2 | 37.5 | 23.2 |
| Rwanda | 3,236 | 35.9 | 85.9 | 70.4 | 61.7 |
| Senegal | 2,327 | 46.6 | 56.8 | 80.6 | 62.4 |
| Sierra Leone | 3,950 | 65.8 | 68.2 | 81.7 | 74.0 |
| South Africa | 1,386 | 61.0 | 75.0 | 84.0 | 78.1 |
| Tanzania | 4,335 | 48.5 | 65.4 | 50.1 | 40.3 |
| Uganda | 5,901 | 42.9 | 61.9 | 52.8 | 37.7 |
| Zambia | 3,905 | 48.4 | 64.0 | 69.5 | 56.3 |
| Zimbabwe | 2,454 | 51.1 | 63.0 | 56.3 | 64.8 |
| Total | 118,614 | 44.3 | 52.6 | 55.4 | 41.6 |

and about 1.2% for newborn PNC services. The overall EC rates are also concerningly low in SSA with average scores of 25.9% for ANC, 38.1% for institutional delivery, 32.1% for maternal PNC, and 24.2% for newborn PNC services (Table 3).

**Selection of cases for further analysis of factors**

We assessed the performance of countries across the four domains of MNH services: ANC, delivery, and PNC for both mothers and newborns as indicated in Table 3. Our analysis revealed varying performance levels among countries in these domains, with Ethiopia consistently ranking the lowest across all domains. Accordingly, Ethiopia and Guinea were selected as bottom-performing countries for ANC, Ethiopia and Nigeria for delivery, and Ethiopia and Angola for PNC. Middle-performing countries included Uganda and Tanzania for ANC and delivery, and Zambia and Benin for PNC. High-performing countries, or top performers, were Ghana and Liberia for ANC, Rwanda and Gabon for delivery, and Gambia and South Africa for PNC. However, health facility and DHS surveys were published in French and/or health facility surveys were not accessible or are not conduced in some countries including Gabon, for example. This has forced us to consider other countries for the analysis. Finaly, we ended up including 10 countries (Liberia, Ghana, Rwanda, South

**Table 3. Countries performance in EC of routine MNH services in Sub-Saharan Africa.**

| Country | 4⁺ANC visits (%) | Country | Delivery care (%) | Country | PNC mothers (%) | Country | PNC newborns (%) |
|---|---|---|---|---|---|---|---|
| Liberia | 63.6 | Rwanda | 80.8 | Gambia | 76.7 | South Africa | 67.9 |
| Ghana | 62.3 | Gabon | 77.6 | Ghana | 73.9 | Ghana | 63.9 |
| Sierra Leone | 52.3 | Burkina Faso | 74.8 | Sierra Leone | 71.1 | Sierra Leone | 61.3 |
| Gabon | 52.0 | Malawi | 72.5 | South Africa | 70.8 | Gabon | 53.7 |
| Gambia | 48.1 | South Africa | 72.1 | Senegal | 65.2 | Gambia | 51.6 |
| South Africa | 45.7 | Sierra Leone | 58.2 | Burkina Faso | 63.3 | Senegal | 51.4 |
| Burkina Faso | 41.0 | Benin | 57.2 | Liberia | 59.9 | Zimbabwe | 47.8 |
| Zimbabwe | 37.4 | Zambia | 55.2 | Kenya | 59.1 | Rwanda | 46.2 |
| Kenya | 32.3 | Ghana | 55.0 | Gabon | 54.2 | Burkina Faso | 43.2 |
| Cameroon | 31.9 | Tanzania | 52.9 | Côte d'Ivoire | 52.2 | Kenya | 42.2 |
| Tanzania | 31.4 | Liberia | 51.6 | Rwanda | 49.6 | Liberia | 41.9 |
| Zambia | 30.5 | Zimbabwe | 51.0 | Zambia | 49.0 | Zambia | 40.8 |
| Angola | 28.2 | Burundi | 49.9 | Benin | 43.4 | Malawi | 31.4 |
| Uganda | 25.8 | Kenya | 48.5 | Cameroon | 34.0 | Côte d'Ivoire | 29.3 |
| Senegal | 25.2 | Uganda | 47.3 | Zimbabwe | 32.1 | Benin | 29.1 |
| Côte d'Ivoire | 24.7 | Senegal | 47.2 | Mali | 30.9 | Cameroon | 27.7 |
| Madagascar | 23.2 | Gambia | 41.4 | Uganda | 29.0 | Tanzania | 21.7 |
| Nigeria | 22.6 | Côte d'Ivoire | 39.4 | Burundi | 25.9 | Uganda | 21.2 |
| Benin | 21.7 | Mauritania | 33.3 | Tanzania | 25.3 | Mali | 18.2 |
| Rwanda | 16.9 | Mozambique | 32.8 | Madagascar | 25.0 | Madagascar | 13.7 |
| Malawi | 16.6 | Cameroon | 31.5 | Guinea | 21.6 | Guinea | 13.3 |
| Mozambique | 16.5 | Mali | 27.9 | Mauritania | 18.3 | Mozambique | 12.4 |
| Burundi | 16.2 | Angola | 15.5 | Malawi | 17.9 | Burundi | 10.5 |
| Mali | 14.4 | Guinea | 14.4 | Nigeria | 16.0 | Mauritania | 9.8 |
| Mauritania | 12.1 | Madagascar | 10.8 | Mozambique | 13.4 | Nigeria | 8.8 |
| Guinea | 9.7 | Ethiopia | 9.1 | Angola | 5.7 | Angola | 3.1 |
| Ethiopia | 7.0 | Nigeria | 9.0 | Ethiopia | 2.7 | Ethiopia | 1.2 |
| **Average** | **25.9** | **Average** | **38.1** | **Average** | **32.1** | **Average** | **24.2** |

Africa, Kenya, Tanzania, Malawi, Mali, Nigeria and Ethiopia) as cases for comparison and factor analysis. The first four countries were selected from top performers. Kenya and Tanzania were selected from medium performers, and the last three are from low-performing countries category in the EC of MNH services. Malawi is among the low performing categories in ANC services and PNC mothers despite being among the high and medium performance categories in institutional delivery and PNC services for newborns, respectively (**See S2 File**).

### Factors that can explain variations in EC of MNH services

**Health system level factors. Availability of antenatal care services.** We have found that the proportion of health facilities providing ANC services and their capacity to deliver those services in terms of availability is consistent with the country's performance in EC of MNH services. For example, the two selected high performing countries in EC of ANC services, Ghana and Liberia, have high percentages of facilities offering ANC services, with 85% and 89% respectively. They also exhibit relatively strong capacity in terms of mean availability of tracer items, at 83% and 84%. However, Tanzania, one of the selected medium performing countries has similar score with the high performing countries. The

lower performing countries had the lowest scores in percentage of facilities offering those services with Ethiopia 80% and Malawi 60%. In addition, their mean availability score of tracer items is 69% and 57%, respectively, indicating a relatively low average capacity to provide comprehensive ANC services across all health facilities in those countries (**Table 4**).

**Service specific readiness for antenatal care.** Ghana, one of the highest performing countries in EC of ANC has better score in availability of trained staff for ANC service delivery than the other countries at 60% (**See** S2 File). Medium-performing countries, Kenya and Tanzania, demonstrated a relative superior capacity with 74% and 72% average readiness scores, respectively, even though the Kenya's readiness score is estimated with lower number of indicators than the other countries due to lack of data about some of the indicators in the trained staff and guidelines domain. The high- and low -performing countries followed those countries with Ethiopia having the lowest average ANC service readiness score at 43% (Fig 2).

**Selected countries capacity for provision of institutional delivery and postnatal care.** We have found significant variations in the mean availability and readiness scores of BEmONC tracer items across the included countries, similar to

**Table 4. The percentage distribution of antenatal care service availability by country.**

| Country | Proportion of facilities offering antenatal care | Tracer indicators of ANC service availability | | | | Mean availability of tracer items |
|---------|--------------------------------------------------|-----------------------------------------------|---|---|---|-----------------------------------|
| | | Iron supplementation | Folic acid supplementation | Tetanus toxoid vaccination | Monitoring for hypertensive disorders of pregnancy | |
| Ghana | 85% | 84% | 84% | 83% | 81% | 83% |
| Liberia | 89% | 86% | 78% | 84% | 85% | 84% |
| Kenya | 81% | 79% | 77% | 48% | 79% | 73% |
| Tanzania | 88% | 76% | 87% | 87% | 87% | 85% |
| Malawi | 60% | 59% | 49% | 59% | 57% | 57% |
| Ethiopia | 80% | 76% | 57% | 74% | 59% | 69% |

**Note:**

Offers antenatal care refers to the proportion of health facilities in those countries that provide antenatal care services.

Mean availability is the overall average score of the mean availability scores of the four items in those health facilities (See S2 File).

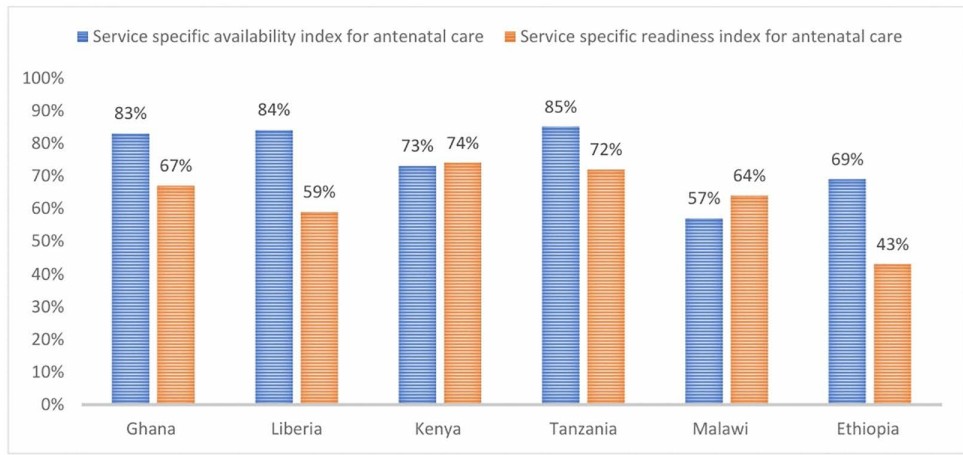

**Fig 2. Service specific availability Vs. service specific readiness index for antenatal care, by country (See S2 File).**

the ANC service indicators. For instance, Malawi and Ghana had the highest mean availability scores for obstetric signal functions offered across the country's health facilities at 82% and 68%, respectively, compared to the medium- and low-performing countries including Ethiopia which stood at 45%. A similar pattern is observed for newborn signal functions, routine perinatal care service indicators, and in the overall mean BEmONC service delivery availability indicator tracer items. Slight variations in the overall mean BEmONC service readiness indicator tracer item scores are also observed across the high, medium and low performing countries (See S2 File). This indicates that the disparities in EC across countries are likely influenced by the variations in both the availability and readiness scores of health services and other enabling societal factors. However, no data was obtained for some countries at all and for some indicators in the Malawi health facility survey. Therefore, mean availability for newborn signal functions, for the tracer items of routine perinatal care and the overall mean availability score of BEmONC tracer items does not apply for the country as indicated in the diagram (Fig 3).

**Community and/or societal level factors.** Disparities in community and/or societal-level factors such as education, household size, media access, and political stability can also play a significant role in determining variations in EC of MNH services even when healthcare service availability and readiness scores are relatively similar across countries. High performing countries had high proportion of women with some or more education than the low performing countries in EC of MNH services. For instance, a wide gap is observed between two high performing countries, south Africa and Ghana, and two low performing countries, Mali and Ethiopia, with the former countries having 88.9% and 70.1% of women attending some secondary and/or higher education, respectively, compared to 21% and 17.2% in the later ones. Mass media and internet access is also higher in those countries. In addition, those positive outlier countries had a relatively better average political stability index than the medium and low performing countries. In contrast, women in medium- and low-performing countries face pronounced barriers from the interplay of these societal factors and MNH service perspectives. For example, the two low performing countries, Ethiopia and Nigeria, had the weakest average political stability index which could be a major factor that could influence infrastructure expansion and health seeking behaviour of women in those countries. In addition, household size and fertility rates are relatively higher in low and medium performing countries than the positive outliers (Table 5).

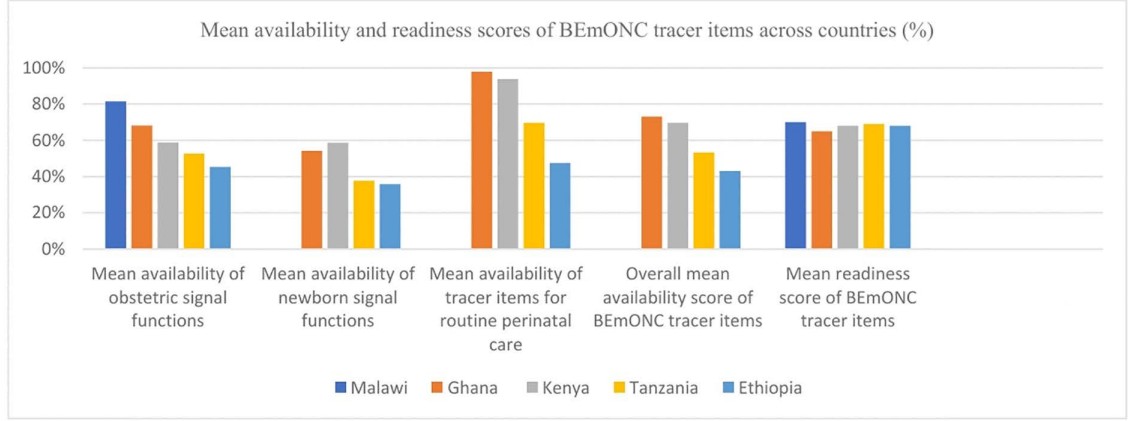

**Fig 3. Percentage distribution of basic emergency and essential obstetric and newborn care service availability and readiness scores by country (See S2 File).**

**Table 5. Distribution of community/societal level factors across the selected countries DHS and The GlobalEconomy.com Data.**

| Country | Proportion of women with some or more school attendance (%) | Proportion of women with primary education (%) | Proportion of women with sone secondary or higher education (%) | Proportion of women employed in the 12 months before surveys (%) | Proportion of women in the middle or more wealth quintiles (%) | Family size/Total fertility rate (average score) | House-hold size (average score) | Proportion of women with internet access (%) | Proportion of women with mass media access (%) | Average political stability index 1996–2023 (−2.5 weak; 2.5 strong) |
|---|---|---|---|---|---|---|---|---|---|---|
| Ghana | 83.9 | 13.8 | 70.1 | 78.2 | 65.6 | 3.9 | 3.5 | 43.3 | 72.7 | −0.02 |
| Liberia | 69.3 | 23.7 | 45.6 | 64.3 | 65.2 | 4.2 | 4.6 | 22.0 | 33.1 | −0.95 |
| Rwanda | 90.6 | 58.3 | 32.3 | 73.3 | 62.4 | 4.1 | 4.3 | 12.3 | 65.6 | −0.52 |
| Malawi | 87.9 | 62.1 | 25.8 | 67.1 | 61.7 | 4.4 | 4.5 | 5.5 | 37.2 | −0.09 |
| South Africa | 98.0 | 9.1 | 88.9 | 38.5 | 60.5 | 2.6 | 3.4 | 47.4 | 82.3 | −0.22 |
| Kenya | 94.5 | 36.3 | 58.1 | 59.7 | 66.7 | 3.4 | 3.7 | 44.2 | 78.5 | −1.15 |
| Tanzania | 83.9 | 53.2 | 30.7 | 64.3 | 66.9 | 4.8 | 4.5 | 12.8 | 45.6 | −0.37 |
| Mali | 34.0 | 13.0 | 21.0 | 61.0 | N/A | 6.3 | 5.8 | N/A | N/A | −0.90 |
| Nigeria | 65.1 | 14.4 | 50.7 | 68.4 | 63.5 | 5.3 | 4.7 | 15.7 | 44.4 | −1.81 |
| Ethiopia | 52.2 | 35.0 | 17.2 | 50.2 | 65.3 | 4.6 | 4.6 | 4.4 | 26.4 | −1.52 |

**Note:** For Details, See Additional File 2.

## Political stability index in Sub Saharan Africa

Political instability can significantly disrupt essential societal systems, including transportation infrastructure, environmental conditions, and community safety, ultimately affecting healthcare access and contributing to widening health inequities [45]. In our analysis, all selected countries have political stability index scores below zero, indicating persistent instability as a major factor for the low EC rates of MNH services across SSA. However, despite the overall instability, high-performing countries tend to exhibit relatively better political stability index scores compared to medium- and low-performing countries, suggesting that even marginal improvements in stability may positively influence healthcare outcomes (**Fig 4**).

## Discussion

We have found that in SSA around 59% of women have completed 4+ANC visits, 73% gave birth in health facilities, and postnatal care coverage within two days was about 58% for mothers and newborns, each. The population average intervention coverage scores for these services range from about 44% for 4+ANC visits to 55% for maternal PNC. We have also found significant disparities in EC of MNH services across SSA countries. Effective coverage rates range from about 7% in Ethiopia to 64% in Liberia for 4+ANC visits, and from 9% in Ethiopia and Nigeria to 81% in Rwanda for institutional delivery. For PNC, the score ranges from about 3% in Ethiopia to 77% in Gambia for mothers and from 1% in Ethiopia to 68% in South Africa for newborns. Some countries like Ghana, Sierra Leone and South Africa have demonstrated high performance consistently across the spectrum of MNH services. In contrast, countries like Nigeria, Angola, and Ethiopia showed the lowest EC rates, particularly for delivery and PNC, with Ethiopia being the least performer in all domains. These differences in performance could be attributed to variations in socioeconomic status and other societal factors [18,46–50]. Furthermore, it could be due to disparities in healthcare access and/or the countries capacity in availability and/or readiness of those services [13,16,17,51].

We have analysed country level societal and health system factors to understand the variations in the EC of MNH services across the countries. At the health system level, we have assessed service-specific availability and readiness indexes of MNH services as they are crucial to understand the health facility capacity of different countries across a spectrum of services. The results indicated that the availability of MNH services has similar patterns with EC estimates of

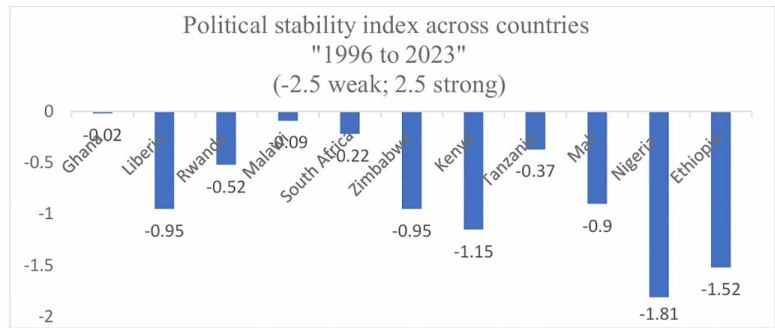

**Fig 4. Average political stability index across selected SSA countries, 1996 to 2023 (See S2 File).**

countries. Overall, the selected high performing countries for comparison with the medium- and low-performing countries had much better capacity in terms of availability of MNH service provision quality indicator tracer items. For instance, Ghana and Liberia, the two positive outliers in EC of 4+ANC visits, had high average proportion scores for tracer indicators of ANC service availability at 83% and 84%, respectively, compared to 43% in Ethiopia and 64% in Malawi, from low performing country categories. Similar pattern is observed between the selected countries EC estimates of MNH services and their health service specific readiness index scores. This indicates that the variation in EC estimates across SSA countries are determined by the country's health facility capacities and other socio-economic, environmental and political factors. Similar findings were reported in previous studies highlighting the significant role of service availability and readiness in determining EC of MNH services. For instance, high EC of ANC services has been found in Palestine in facilities having adequate resources, such as labs and ultrasounds, compared to their counterparts [51].

On the negative side, facility readiness limitations including staff shortages, limited test kits, and inconsistent practices, are reported as common bottlenecks for low EC of MNH services. For example, health facility readiness and clinical practice gaps reduced EC for syphilis and pre-eclampsia screenings by over 50% in certain districts, in Tanzania [16]. Similarly, human resource shortages have been identified as a significant barrier to EC in HIV counselling and malaria presumptive treatment during pregnancy [17]. These findings indicate that even though service availability and readiness did not provide guarantee for the delivery of quality services, they are prerequisites for service quality and play a significant role in explaining the variations in EC of MNH services in SSA. Furthermore, the findings have indicated the critical need for improvements in both health service availability and readiness scores across countries in SSA to enhance MNH care, particularly in low performing countries like Ethiopia [52]. One of the major determinants for variations in service availability and readiness is health care financing. For instance, there is huge gap in current health expenditure per capita between Ethiopia and South Africa with average scores for 2 years (2013 and 2014) before the countries DHS being US$20.5 and US$515.5, respectively (See S2 File).

Nevertheless, those health system-level factors alone cannot fully explain the variations in EC of MNH services across countries. In addition, availability is used to refer simple physical presence of various healthcare service tracer indicators and does not encompass more complex factors like geographical barriers, travel time, and/or user behaviour [53]. Societal factors, such as education, employment, family size and political stability are equally significant. High-and medium performing counties are better positioned from these factors perspectives which could have positively influenced the health seeking behaviour of women and service coverage outcomes. For instance, in South Africa and Rwanda, 98% and 90.6% of women have some education or more school attendance, and about 89% and 32% have attended and/or completed secondary education or higher, respectively, which could contribute to better health literacy and utilization of services in those countries [54,55]. Mass media and internet access is also better in high performing countries like South Africa

(82.3% and 47.4%) [54] and Ghana (72.7% and 43.3%) [56] which facilitates the ease dissemination of health-related information to the general public [23]. Furthermore, those countries also have relatively small family size and better political stability index. For instance, the average fertility rate in South Africa is 2.6 according to the country's 2016 survey [54], which is much lower compared to the low performing countries: Ethiopia (4.6) [23], Nigeria (5.3) [57] and Mali (6.3) [58].

Among the middle performing countries, Kenya has stronger scores in different societal factors and even much higher than some of the high performing countries, in some situations. For instance, about 95% of women have some education or more school attendance, and 58% have attended and/or completed secondary or higher education. It also has better mass media and internet access as the high performing countries with scores being 78.5% and 44.2%, respectively [24]. However, it has very weak average political stability index in the past decades next to Nigeria and Ethiopia [40]. Societal factors have demonstrated the existence of more complex challenges that could reduce the potentials of accessing service for MNH care, particularly in low performing countries. For example, in Mali, only 34% of reproductive age group women have some or more education, and the proportion who have attended and/or completed secondary education or higher is only 21%. The country also has the largest total fertility rate and household sizes with average scores of 6.3 and 5.8, respectively, according to the country's 2018 DHS report [58].

Ethiopia and Nigeria also share similar complex futures like Mali including having weak average political stability index scores according to TheGlobalEconomy.com data [40]. In Ethiopia, the lowest performing country in EC of MNH services, about half (52.2%) of women have some education or more school attendance. However, only 17.2% have attended and/or completed secondary education or higher. In addition, only 4.4% and 26.4% of women have internet and mass media access, respectively, which might potentially be hindering healthcare information access for MNH services [23]. These factors can make healthcare access more challenging in those countries. The proof for this statement is that the proportion of women having higher MNH service contact coverage rates is generally higher across the high performing countries in EC than the medium and low performing countries. For instance, there is a wide gap in 4+ANC visit coverage rates between Ethiopia and Ghana (33% Vs. 88%), and in coverage rates of institutional delivery between Ethiopia and South Africa (39% Vs. 98%).

Other studies have reported consistent findings to our study findings across the literature in that women's socioeconomic status is linked to variations in the EC of MNH services [18,46–50]. For instance, differences in educational status are reported as one of the main reasons for disparities in EC [47–50]. Women's occupation [49], and place of residence have also been reported as having an association with variations in EC of ANC services [49,59]. Furthermore, wealth index has been cited as a dominant factor for variations in EC of MNH services [14,48]. These findings suggest that significant improvements in the EC of MNH services, particularly in low-performing countries, are unlikely without broader societal progress and a stable political system. Political instability can hinder healthcare investments, disrupt service delivery, and weaken institutional capacity, making it difficult to expand health services and strengthen existing facilities [60,61]. Therefore, addressing these challenges requires a comprehensive approach that integrates health policies with broader political and socioeconomic reforms to ensure sustainable and equitable improvements in MNH service coverage outcomes.

## Strength and limitations

This study has certain limitations. Firstly, the availability and readiness tracer indicators for MNH, particularly for delivery and postnatal care, lack specificity in the health facility surveys of the included countries. Having separate indicators for routine institutional delivery and postnatal care would have provided more detailed insights. Additionally, the absence of health facility survey data for some countries necessitated the inclusion of other countries in the factor analysis, even though they were not initially selected as high, medium, or low performers. Furthermore, the lack of comprehensive data limits the examination of factors such as governance, leadership, and communication systems across these countries. Despite these limitations, the study has several notable strengths. One key strength is the use of representative samples from both the DHS for women aged 15–49 years and the health facility survey's, ensuring comparability across countries.

Moreover, the standardized data collection methods employed by the DHS program, along with the concurrent timing of health facility surveys and DHS data collection, enhance the reliability of the findings. The study has also highlighted the importance of understanding the multilevel factors influencing the EC of MNH services across sub-Saharan African countries. It underscores that both service availability and readiness, alongside societal factors, play a crucial role in shaping the EC of MNH services.

## Conclusion

There is huge variation in EC of MNH services across countries in SSA. These discrepancies are highly likely influenced by multilevel factors including the health system and societal factors. Countries in the high-performing category in EC of MNH services have better service specific availability and readiness scores for MNH care than the medium- and low-performing countries. In addition, current health expenditure per capita is much higher in these countries compared to the low performing countries. They also have favourable societal factors including high proportion of women attending primary and/or more school levels, better mass media and internet access, and relatively lower political instability index. Medium performing countries also share similar patterns with their performance category in terms of health service availability and readiness indicator scores, and the societal factors. Low-performing countries like Ethiopia and Nigeria presented complex futures including having low health service availability and readiness scores. They have also demonstrated unfavourable societal factors like low educations status, internet and mass media access, and very weak political stability index that hinders the utilization and delivery of MNH services.

Sub-Saharan Africa countries, particularly low performers in EC of MNH services, need to learn from the positive outliers in order to enhance their country's health service utilization and delivery capacity. For instance, despite the challenges such as low women education and political instability, which may take years to address, countries can take targeted actions over the next five years to improve EC. This includes not only building new health facilities for expansion but also enhancing the capacity of existing ones by ensuring better staffing, training, and access to essential equipment, medications, and supplies. Other key strategies could include implementing community-based health education and outreach programs, leveraging digital and mobile health technologies to reach underserved populations, and establishing robust monitoring and accountability mechanisms. The countries also need to understand that addressing these challenges requires a comprehensive approach that integrates health policies with broader political and socioeconomic reforms to ensure sustainable and equitable improvements in MNH service coverage outcomes.

## Supporting information

**S1 File. Estimating EC for MNH visits and quality indicators.**
(DOCX)

**S2 File. Extracted data on health system and societal-level factors among cases.**
(DOCX)

**S3 File. Extracted Mini-Dataset and graphic outputs from DHS Datasets and reviewed documents.**
(XLS)

## Author contributions

**Conceptualization:** Ayelign Mengesha Kassie, Yibeltal Assefa.

**Data curation:** Ayelign Mengesha Kassie, Solomon Woldeyohannes, Anteneh Zewdie, Eskinder Wolka, Yibeltal Assefa.

**Formal analysis:** Ayelign Mengesha Kassie, Solomon Woldeyohannes, Anteneh Zewdie, Eskinder Wolka, Yibeltal Assefa.

**Investigation:** Ayelign Mengesha Kassie, Solomon Woldeyohannes, Anteneh Zewdie, Eskinder Wolka, Yibeltal Assefa.

**Methodology:** Ayelign Mengesha Kassie, Solomon Woldeyohannes, Anteneh Zewdie, Eskinder Wolka, Yibeltal Assefa.

**Supervision:** Solomon Woldeyohannes, Anteneh Zewdie, Eskinder Wolka, Yibeltal Assefa.

**Validation:** Ayelign Mengesha Kassie, Solomon Woldeyohannes, Anteneh Zewdie, Eskinder Wolka, Yibeltal Assefa.

**Visualization:** Ayelign Mengesha Kassie, Solomon Woldeyohannes, Anteneh Zewdie, Eskinder Wolka, Yibeltal Assefa.

**Writing – original draft:** Ayelign Mengesha Kassie.

**Writing – review & editing:** Ayelign Mengesha Kassie, Solomon Woldeyohannes, Anteneh Zewdie, Eskinder Wolka, Yibeltal Assefa.

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
