## [Decision Letter · Decision Letter 0]

20 Nov 2025

PONE-D-25-40440Effective Coverage of Maternal and Newborn Health Services in Sub-Saharan Africa: What Distinguishes High from Medium and Low Performers?PLOS ONE?

Dear Dr. Kassie,

Thank you for submitting your manuscript to PLOS ONE. After careful consideration, we feel that it has merit but does not fully meet PLOS ONE’s publication criteria as it currently stands. Therefore, we invite you to submit a revised version of the manuscript that addresses the points raised during the review process.

We look forward to receiving your revised manuscript.

Kind regards,

Abu Sayeed, MSc

Academic Editor

PLOS ONE

Journal Requirements:

3. Please ensure that you refer to Figure 4 in your text as, if accepted, production will need this reference to link the reader to the figure.

4. We note you have included a table to which you do not refer in the text of your manuscript. Please ensure that you refer to Table 5 in your text; if accepted, production will need this reference to link the reader to the Table.

5. Please include a copy of Table 6 which you refer to in your text on page 14.

Reviewers' comments:

Reviewer's Responses to Questions

**Comments to the Author**

1. Is the manuscript technically sound, and do the data support the conclusions?

Reviewer #1: Yes

Reviewer #2: Yes

Reviewer #3: Partly

2. Has the statistical analysis been performed appropriately and rigorously?

Reviewer #1: Yes

Reviewer #2: Yes

Reviewer #3: Yes

3. Have the authors made all data underlying the findings in their manuscript fully available?

Reviewer #1: Yes

Reviewer #2: Yes

Reviewer #3: No

4. Is the manuscript presented in an intelligible fashion and written in standard English?

Reviewer #1: Yes

Reviewer #2: Yes

Reviewer #3: Yes

Reviewer #1: The authors have selected an important topic for discussion. Effective coverage is key to improving MNH outcomes. One suggestion is to define effective coverage in the abstract and at the beginning of the paper, so readers understand the term right from the beginning. The analysis is quite complex and one can get lost in what the authors are referring to - for example intervention coverage versus effective coverage. Please clarify wherever possible or maybe refer to an intervention bundle?

Table 3- not quite sure what the different colors were indicating. Please specify. On page 23 where the authors talk about "facility readiness, staff shortages, availability of test kits" - what is the source of that data?

In the conclusion section- it would be good if the authors can specify what countries can do in the next 5 years despite low educational level and existing political instability. Are there some means to increase effective coverage despite some of these background constraints that may take years to change

Reviewer #2: The study is well designed, and the article well written. This research adds important information to the body of knowledge in this area of study. I commend the authors on their work and look forward to reading and sharing the published version.

Reviewer #3: 1. Ensuring that the methods are described with sufficient detail, including sample sizes, participant selection criteria, data collection instruments, and procedures.

2. Providing comprehensive descriptions of the analytical techniques used, including statistical tests, assumptions checked, and handling of confounding variables.

3. Discuss the paper with other similar literature findings from Europe and Asia.

Explain in detail what each paragraph of the discussion means.

Connecting the findings to existing literature, especially on the factors affecting maternal and neonatal health services, and discussing practical or policy implications.

4. Providing accessible data in accordance with the journal’s policies will improve transparency.

.

Reviewer #1: No

Reviewer #2: **Yes:**Magdeline AagardMagdeline AagardMagdeline AagardMagdeline Aagard

Reviewer #3: No

---

## [Author Response · Author response to Decision Letter 1]

25 Nov 2025

Point by Point Response

PONE-D-25-40440

Effective Coverage of Maternal and Newborn Health Services in Sub-Saharan Africa: What Distinguishes High from Medium and Low Performers?

PLOS ONE

Editors’ comments:

Thank you for submitting your manuscript to PLOS ONE. After careful consideration, we feel that it has merit but does not fully meet PLOS ONE’s publication criteria as it currently stands. Therefore, we invite you to submit a revised version of the manuscript that addresses the points raised during the review process.

We look forward to receiving your revised manuscript.

Kind regards,

Abu Sayeed, MSc

Academic Editor

PLOS ONE

Author’s response:

Dear Editor(s),

We thank you very much for inviting us to submit a revised version of the manuscript that addresses the points raised during the review process. We greatly appreciate the reviewers and editor’s comments, which have helped us improve the clarity and quality of our work. We have carefully addressed all the points raised during the review process.

Journal Requirements:

Author’s response:

We have carefully reviewed the manuscript to ensure that it fully complies with PLOS ONE’s style requirements, including formatting, structure, and file naming conventions. All necessary adjustments have been made accordingly.

Author’s response:

Ethics statement is included in the Methods section of the manuscript only. Located at the end of operational definition and variable section.

3. Please ensure that you refer to Figure 4 in your text as, if accepted, production will need this reference to link the reader to the figure.

Author’s response:

We have carefully revised the manuscript to ensure that all figures, including Figure 4, are properly cited in the text.

4. We note you have included a table to which you do not refer in the text of your manuscript. Please ensure that you refer to Table 5 in your text; if accepted, production will need this reference to link the reader to the Table.

Author’s response:

Thank you for indicating the citation error. Table 5 is now properly cited in the text.

5. Please include a copy of Table 6 which you refer to in your text on page 14.

Author’s response:

The reference to Table 6 on page 14 was incorrect and has been corrected to refer to Table 5. The manuscript now correctly cites Table 5, and all table references have been carefully reviewed for accuracy.

Author’s response:

We have included captions for our Supporting Information files at the end of the manuscript and updated some in-text citations to match accordingly.

Additional file 1. Estimating EC for MNH visits and quality indicators.

Additional file 2. Extracted data on health system and societal-level factors among cases.

Additional file 3. Extracted Mini-Dataset and graphic outputs from DHS Datasets and reviewed documents.

Author’s response:

No specific publications were recommended by the reviewers other than the general suggestions to revise the manuscript and consider additional studies for comparison and analysis. The manuscript has been revised accordingly.

Author response:

No retracted articles were cited in our manuscript. We have carefully reviewed the reference list to ensure that it is complete and accurate.

Reviewers' comments:

Reviewer's Responses to Questions

Comments to the Author

1. Is the manuscript technically sound, and do the data support the conclusions?

Reviewer #1: Yes

Reviewer #2: Yes

Reviewer #3: Partly

Author’s response:

Dear reviewers,

Thank you so much for confirming that the manuscript describes a technically sound piece of scientific research.

2. Has the statistical analysis been performed appropriately and rigorously?

Reviewer #1: Yes

Reviewer #2: Yes

Reviewer #3: Yes

Author’s response:

We believe so. Thank you so much!

3. Have the authors made all data underlying the findings in their manuscript fully available?

The PLOS Data policy requires authors to make all data underlying the findings described in their manuscript fully available without restriction, with rare exception (please refer to the Data Availability Statement in the manuscript PDF file). The data should be provided as part of the manuscript or its supporting information or deposited to a public repository. For example, in addition to summary statistics, the data points behind means, medians and variance measures should be available. If there are restrictions on publicly sharing data—e.g. participant privacy or use of data from a third party—those must be specified.

Reviewer #1: Yes

Reviewer #2: Yes

Reviewer #3: No

Author’s response:

Thank you so much. we have made the data underlying the findings in our manuscript fully available. The relevant data are provided in the following supplementary files:

• Additional File 2: Extracted data on health system and societal-level factors among cases.

• Additional File 3: Extracted mini-dataset and associated graphic outputs derived from the DHS datasets.

However, the full DHS datasets cannot be shared directly, as they are accessible only through the DHS Program. Researchers can obtain them upon request from the DHS website in accordance with their data access policies.

4. Is the manuscript presented in an intelligible fashion and written in standard English?

Reviewer #1: Yes

Reviewer #2: Yes

Reviewer #3: Yes

Author’s response:

We have proofread the manuscript and believe it is suitable for publication in its current version.

5. Reviewer Specific Comments to the Author

Reviewer #1: The authors have selected an important topic for discussion. Effective coverage is key to improving MNH outcomes.

1. One suggestion is to define effective coverage in the abstract and at the beginning of the paper, so readers understand the term right from the beginning. The analysis is quite complex and one can get lost in what the authors are referring to - for example intervention coverage versus effective coverage.

Author’s response:

Dear reviewer,

We thank you very much for highlighting this important point. We have included the definition of effective coverage both in the abstract and the introduction sections to ensure that readers understand the term from the outset. In addition, we have clarified distinctions between intervention coverage and effective coverage in the manuscript’s methodology section particularly.

2. Please clarify wherever possible or maybe refer to an intervention bundle?

Author’s response:

Thank you. In our study, intervention coverage was estimated using a set of service-specific quality indicators for each MNH service domain, which can be considered an “intervention bundle.” The specific indicators included in each bundle are provided in Additional file 1. For example, the 4+ ANC bundle included 9–10 items (depending on country), institutional delivery included 4 items, and postnatal care for mothers and newborns included 3 and 8 items, respectively. Each bundle reflects the recommended services that women and newborns should receive to achieve the intended health benefit.

3. Table 3- not quite sure what the different colours were indicating. Please specify.

Author’s response:

Thank you, so much dear reviewer. In Table 3, colours were used to indicate relative effective coverage performance across MNH service domains using a tertile classification method. Green represents countries in the highest tertile, yellow represents the middle tertile, and red represents the lowest tertile. This approach helps us illustrate the disparities across SSA countries. For example, Ghana and Liberia had the highest effective coverage for four or more ANC visits, Rwanda had the highest institutional delivery, and Gambia and South Africa led in maternal and newborn postnatal care. In contrast, countries such as Ethiopia and Nigeria had the lowest effective coverage scores. Detail descriptions have been provided in methods sections.

4. On page 23 where the authors talk about "facility readiness, staff shortages, availability of test kits" - what is the source of that data?

Author’s response:

Dear reviewer,

The data on facility readiness, staff availability, and the availability of essential test kits and supplies were primarily obtained from countries’ health facility surveys and their summary reports (for instance, countries service availability and readiness assessment, and service provision assessments). When available, we supplemented these data with information from global databases, including the Global Health Observatory repository and other publicly accessible sources, to capture indicators such as core health worker density and service inputs. Facility readiness scores were calculated for facilities providing specific services, considering key staff, essential equipment, medicines, diagnostics, and infrastructure. Availability refers to the proportion of facilities offering the service and having the necessary tracer items. These measures provide a harmonized, cross-country assessment of health system capacity for MNH services (details are provided under Additional file 2).

5. In the conclusion section- it would be good if the authors can specify what countries can do in the next 5 years despite low educational level and existing political instability. Are there some means to increase effective coverage despite some of these background constraints that may take years to change

Author’s response:

Dear reviewer,

We thank you very much for the insightful comments. We have included these points in the conclusion section. Despite challenges such as low educational levels and political instability that could take years to fix, countries, particularly low performing ones can take targeted actions over the next five years to increase effective coverage of maternal and newborn health services. Some of the key strategies could include strengthening the core health workforce through targeted training and task-shifting, ensuring facility readiness with essential medicines, equipment, and diagnostics, and implementing community-based outreach programs to raise awareness and demand for services. Leveraging mobile and digital health technologies can also help reach underserved populations.

Reviewer #2: The study is well designed, and the article well written. This research adds important information to the body of knowledge in this area of study. I commend the authors on their work and look forward to reading and sharing the published version.

Author’s response:

Dear reviewer,

We thank you for your positive and encouraging comments. We appreciate the recognition of the study’s design and contribution, and we are grateful for the support and encouragement.

Reviewer #3:

1. Ensuring that the methods are described with sufficient detail, including sample sizes, participant selection criteria, data collection instruments, and procedures. Providing comprehensive descriptions of the analytical techniques used, including statistical tests, assumptions checked, and handling of confounding variables.

Author’s response:

Dear reviewer,

We thank you for the valuable comments. We have provided detailed descriptions of our methods, including the study design, sample size, participant selection criteria, and data sources. We used a mixed-methods case study to examine societal and health system factors affecting effective coverage of MNH services in 27 Sub-Saharan African countries. A weighted sample of 118,614 women who delivered a live newborn within two years prior to DHS surveys in 27 Sub-Saharan African countries was analysed. Data collection instruments are standardized in the DHS program and are applicable for every country. For, the documents review, Countries health facility survey summary reports, DHS summary reports, and global databases (Global health observatory repository, Global Health Expenditure Database, TheGlobalEconomic.com) were assessed. Analytical techniques for estimating effective coverage (EC) of maternal and newborn health services included calculation of contact coverage, averaging service-specific quality scores, and combining them to derive EC rates. All analyses accounted for survey weights, and relevant assumptions were addressed in assessing countries by performance.

3. Discuss the paper with other similar literature findings from Europe and Asia. Explain in detail what each paragraph of the discussion means. Connecting the findings to existing literature, especially on the factors affecting maternal and neonatal health services and discussing practical or policy implications.

Author’s response:

We thank the reviewer for this insightful suggestion. We have expanded the discussion on factors affecting maternal and neonatal health services and elaborated on the practical and policy implications of our findings. Our analysis primarily focuses on case studies of societal and health system factors that distinguish high-performing countries from medium- and low-performing countries. Accordingly, the discussion emphasizes the findings and implications from this perspective. However, we have also incorporated comparisons with relevant studies from other Multinational and regional studies to situate our results within the broader literature while keeping the discussion focused.

4. Providing accessible data in accordance with the journal’s policies will improve transparency.

Author’s response:

We have uploaded all data in accordance with the journal’s policies to ensure transparency and facilitate reproducibility.

5. PLOS authors have the option to publish the peer review history of their article (what does this mean?). If published, this will include your full peer review and any attached files. If you choose “no”, your identity will remain anonymous, but your review may still be made public.

Author’s response:

Dear editor (s),

We acknowledge the option to publish the peer review history. We are comfortable with the review being made public, including the full peer review and any attached files.

---

## [Decision Letter · Decision Letter 1]

30 Mar 2026

Effective Coverage of Maternal and Newborn Health Services in Sub-Saharan Africa: What Distinguishes High from Medium and Low Performers?

PONE-D-25-40440R1

Dear Dr. Kassie,

We’re pleased to inform you that your manuscript has been judged scientifically suitable for publication and will be formally accepted for publication once it meets all outstanding technical requirements.

Kind regards,

Marianne Clemence

Staff Editor

PLOS One

Additional Editor Comments (optional):

Reviewers' comments:

Reviewer's Responses to Questions

**Comments to the Author**

Reviewer #2: All comments have been addressed

Reviewer #3: (No Response)

2. Is the manuscript technically sound, and do the data support the conclusions?

Reviewer #2: Yes

Reviewer #3: (No Response)

3. Has the statistical analysis been performed appropriately and rigorously?

Reviewer #2: Yes

Reviewer #3: (No Response)

4. Have the authors made all data underlying the findings in their manuscript fully available?

Reviewer #2: Yes

Reviewer #3: (No Response)

5. Is the manuscript presented in an intelligible fashion and written in standard English?

Reviewer #2: Yes

Reviewer #3: (No Response)

Reviewer #2: Thank you for addressing each of the revisions clearly and succinctly. I look forward to seeing your article in publication.

Reviewer #3: (No Response)

.

Reviewer #2: **Yes:**Magdeline Aagard, EdD, MBA, BANMagdeline Aagard, EdD, MBA, BANMagdeline Aagard, EdD, MBA, BANMagdeline Aagard, EdD, MBA, BAN

Reviewer #3: No

---

## [Editor Report · Acceptance letter]

PONE-D-25-40440R1

PLOS One

Dear Dr. Kassie,

I'm pleased to inform you that your manuscript has been deemed suitable for publication in PLOS One. Congratulations! Your manuscript is now being handed over to our production team.

Kind regards,

on behalf of

Dr Marianne Clemence

Staff Editor

PLOS One